# Actively Learning Costly Reward Functions for Reinforcement Learning

## Abstract

Transfer of recent advances in deep reinforcement learning to real-world applications is hindered by high data demands and thus low efficiency and scalability. Through independent improvements of components such as replay buffers or more stable learning algorithms, and through massively distributed systems, training time could be reduced from several days to several hours for standard benchmark tasks. However, while rewards in simulated environments are well-defined and easy to compute, reward evaluation becomes the bottleneck in many real-world environments, e.g., in molecular optimization tasks, where computationally demanding simulations or even experiments are required to evaluate states and to quantify rewards. When ground-truth evaluations become orders of magnitude more expensive than in research scenarios, direct transfer of recent advances would require massive amounts of scale, just for evaluating rewards rather than training the models. We propose to alleviate this problem by replacing costly ground-truth rewards with rewards modeled by neural networks, counteracting non-stationarity of state and reward distributions during training with an active learning component. We demonstrate that using our proposed ACRL method (**a**ctively learning **c**ostly rewards for **r**einforcement **l**earning), it is possible to train agents in complex real-world environments orders of magnitudes faster. By enabling the application of reinforcement learning methods to new domains, we show that we can find interesting and non-trivial solutions to real-world optimization problems in chemistry, materials science and engineering.

## 1 Introduction

Reinforcement Learning (RL) techniques have achieved impressive results in a wide range of applications such as robotics (Kober et al., 2013), games (Mnih et al., 2015; Silver et al., 2016; Vinyals et al., 2019) or natural sciences (Mahmud et al., 2018; Zhou et al., 2019). This success is the result of improvements along multiple independent branches of RL research such as an improved understanding of rewards in difficult environments (Schaal, 1997; Abbeel & Ng, 2004; Christiano et al., 2017; Wirth et al., 2017), more sample-efficient training via experience replay (Lin, 1992; Schaul et al., 2015; Andrychowicz et al., 2017; Kong et al., 2021) or more effective sampling via active learning (Daniel et al., 2015; Cui & Niekum, 2018; Biyik et al., 2020), more powerful algorithms (Mnih et al., 2015; Van Hasselt et al., 2016; Lillicrap et al., 2015; Fujimoto et al., 2018; Haarnoja et al., 2018a) and more efficient and scalable implementations (Horgan et al., 2018; Dalton & frosio, 2020; Hessel et al., 2018) of established techniques. These extensions were primarily developed and benchmarked in simulated environments such as OpenAI Gym (Brockman et al., 2016) or MuJoCo (Todorov et al., 2012), where rewards are well-defined and computationally cheap to obtain.

However, in real-world tasks rewards may be either difficult to formulate or to collect. There has been extensive work on how to formulate and quantify rewards in scenarios where agents have to learn from demonstrations (Schaal, 1997; Abbeel & Ng, 2004) or from ranked alternatives (Christiano et al., 2017; Wirth et al., 2017). These methods are mainly developed within the field of robotics, where feedback frequently is provided by a human supervisor. Thus, since human feedback is relatively expensive, it is desirable to reduce the number of expert evaluations. Active reward learning techniques aim to reduce the number of expert queries by selecting only the most informative ones, usually employing uncertainty measures within

the Bayesian framework (Daniel et al., 2015; Cui & Niekum, 2018; Biyik et al., 2020; Lindner et al., 2021) as a selection criterion.

Existing literature focuses on developments in simulated environments and real-world tasks in fields such as robotics. While in the former case rewards are clearly formulated and cheap to obtain, in the latter case rewards are typically difficult to formulate and/or quantify, e.g., in object manipulation tasks (Jangir et al., 2020). However, in many other fields rewards appear to have different properties than in these scenarios, for which most of existing work has been done. In a natural sciences and engineering context, for example, rewards are frequently the result of computationally demanding optimization procedures or algorithms. The current trend in reinforcement learning is to tackle these issues with massive scale by running multiple independent copies of agent-environment interactions in parallel. While using massive amounts of compute resources may be justified by the outstanding results such as (Silver et al., 2017; Jumper et al., 2021), following this trend in scenarios with complex reward evaluations has two major drawbacks. First, it may exclude all but the largest institutions to engage in this area of research at all. Second, establishing scale as the default to gather enough data to train agents can be considered *Red AI* (Schwartz et al., 2020).

In contrast to a large body of existing work on optimizing with reinforcement learning, we explicitly focus on training with costly rewards, whereas prior work mostly reported results on cheap or approximate quantities (Zhou et al., 2019; Goel et al., 2021; Bhola et al., 2023). When ground-truth evaluations become orders of magnitude more expensive than in these scenarios, direct transfer of these methods would require massive amounts of scale, just for evaluating rewards rather than training the models. In this paper, we develop a framework combining reinforcement and active learning to resolve the issue of reward collection in real-world scenarios, where the relevant domain-specific quantities are difficult to obtain. We show that within our framework, which we term **ACRL**[1], neural networks, pre-trained on a relatively small initial dataset and regularly updated during training via an active learning approach, can be used as reward proxies and that agents trained within this framework achieve competitive results across different real-world tasks with varying computational cost.

## 2 Related work

### 2.1 Learning reward functions

In theory, every agent accumulates rewards under a unified mathematical framework. In practice, though, the exact properties of a reward function depend on the task. For example, rewards can be immediate or delayed and the reward signal can be binary, discrete or real. In simulated environments like OpenAI Gym and MuJoCo rewards are well-defined and exposed to the agent via a simulator interface. In fields like robotics, rewards can become complex, high-level signals of desired behavior, e.g., to manipulate an object in a particular manner (Jangir et al., 2020). Since the formulation of a reward function is often difficult in the latter case, early work (Schaal, 1997; Abbeel & Ng, 2004) aimed to infer an unknown reward function solely from demonstration. While alleviating the issue of reward formulation, demonstrations by a human supervisor are costly to obtain. As an alternative, preference-based learning (Christiano et al., 2017; Wirth et al., 2017) allows feedback to be a relative preference over a set of trajectories rather than a quantitative measure of goodness.

In our work, we focus on problems where reward functions correspond to quantities such as evaluations of properties of physical systems. For example, this can be expensive quantum-mechanical simulations for evaluation of molecular properties. Given such a reward function, we can formulate an optimization process as goal-directed search within the reinforcement learning framework, in which states of high rewards correspond to more optimal solution spaces of the underlying problem and vice versa. We therefore propose to replace the ground-truth reward function with an approximate model and to jointly train it with the agent to account for non-stationarity of state and reward distributions during exploration. Due to their ability to generalize, our agents are able to solve optimization tasks with varying constraints, which is, in general, not trivially doable using conventional optimization and search methods.

---

[1]Our code is available at: `https://github.com/32af3611/acrl`

## 2.2 Active reward learning and sample efficiency

Active reward learning techniques (Daniel et al., 2015; Cui & Niekum, 2018; Biyik et al., 2020) build upon the insight that not all training samples are equally important for learning and aim to select only those samples which are most beneficial for learning. The selection is usually done by some form of uncertainty estimation, often within the Bayesian framework. Reducing the number of state queries is vital in cases where reward evaluation is expensive. While existing work employs active learning to reduce the number of queries for the agent to accelerate convergence of the RL training (Lindner et al., 2021), we employ active learning for the reward model such that predictions become more accurate on states the agent visits during exploration.

In vanilla RL, every observation is used only once to update the agent's policy, making learning slow and sample-inefficient. A popular technique to overcome this is to use *experience replay* (Lin, 1992) in off-policy algorithms, which improves sample-efficiency in terms of sample usage by storing experience in a *replay buffer* and performing parameter updates on batches uniformly sampled from it. Improvements of experience replay use different forms of non-uniform sampling (Schaul et al., 2015; Kong et al., 2021), handle sparse and binary reward signals and multi-goal environments (Andrychowicz et al., 2017), and are also extended to a distributed context (Horgan et al., 2018).

In our work, we do not aim to increase sample-efficiency of the RL training process. Rather, we avoid expensive ground-truth evaluations for known regions of the state space by using a reward model. We increase the size of this region over the course of training by providing ground-truth labels for a small fraction of states selected by some sampling method. Since our framework assumes a modification of the environment's properties, i.e., the reward, it is possible to use techniques like *IDRL* with our framework to reduce the number of reward model queries. Specifically, we show that we can improve training time without using any of the more advanced techniques in the reinforcement learning toolbox to keep our framework lean and to avoid common reproducibility issues (Huang et al., 2022; Henderson et al., 2018).

## 2.3 Efficient implementations

The effects of other extensions within the RL framework have been studied in Hessel et al. (2018), showing recent advances can be integrated to improve their standalone-performance. From a practical point of view, the authors of Stooke & Abbeel (2018) provide a unified implementation view of RL algorithms to leverage modern, parallel hardware architectures to further reduce training time.

In our work, we do not aim to leverage massively scaled reinforcement learning in order to solve the issue of costly rewards. Rather, we propose an extension to restore the effectiveness of these methods in scenarios where their efficiency would be threatened by the reward evaluation bottleneck. We note that our method is scalable and naturally can be integrated into distributed architectures such as those in Horgan et al. (2018).

## 2.4 Reinforcement Learning for optimization

The idea of optimizing functions with reinforcement learning was already investigated several decades ago (Williams & Peng, 1991). In (Williams & Peng, 1991), the authors used REINFORCE (Williams, 1992)-like algorithms on a set of experiments with known maxima to show that it is possible to learn an adaptive system that generates optima by trial-and-error. Interestingly, they found that their algorithms converge to suboptimal single-mode solutions in the presence of multiple equally-valued actions and corrected this behavior by maximizing entropy in addition to reward, which today is a standard technique in robust reinforcement learning and also a core component in one of the current state-of-the-art algorithms, *Soft Actor-Critic* and its variants (Haarnoja et al., 2018a;b; Christodoulou, 2019).

Since then, reinforcement learning has been applied to many instances of combinatorial optimization due to its ability to efficiently explore large spaces without handcrafted heuristics. Canonical NP-hard problems such as the *Travelling Salesman Problem* (TSP) and other graph-related problems have been the focus due to the difficulty of obtaining optimal solutions for these problems (Bello et al., 2016; Khalil et al., 2017). Furthermore, there is an increased interest in using these methods in real-world applications, with applications

for road (Yu et al., 2019) or computer (Vesselinova et al., 2020) networks. A broader overview of machine learning for combinatorial optimization can be found in (Bengio et al., 2021).

Besides applications in computer science, reinforcement learning has been applied for optimization and discovery in a variety of other domains. In chemistry and materials science, optimizing properties of molecules or their molecular graphs, respectively, is of major interest. Existing work such as *MolDQN* (Zhou et al., 2019) uses reinforcement learning to find local modifications of molecules that yield improved properties. Besides single-property optimizations, other work aims for molecules meeting multiple criteria at the same time (Goel et al., 2021), geometry optimization (Ahuja et al., 2021) or design and discovery (Fromer & Coley, 2023; Olivecrona et al., 2017; Pereira et al., 2021).

Another source of interest is the optimization of airfoils to improve their aerodynamic properties, with all kinds of applications in aeronautics. While traditionally these kinds of problems are tackled with optimization methods such as gradient-based optimization, the authors in (Dussauge et al., 2023) argue that these methods, even though computationally efficient in large spaces, are susceptible to poor local minima and do not work well with non-linear cost functions. While machine learning techniques are less susceptible to these kinds of errors, the authors in (Bhola et al., 2023) point out that using high-fidelity data for training can become prohibitively expensive.

While there is much interest in using reinforcement learning for different kinds of optimization problems, in many cases, most effort is spent on finding a solution, with the general assumption that it can be verified fast. Methods such as *MolDQN* (Zhou et al., 2019) or *MoleGuLAR* (Goel et al., 2021) optimize cheap properties, e.g., *logP* and *QED*, while in the case of airfoil design, lower-fidelity data is used to accelerate data generation (Bhola et al., 2023). For real-world domains, the validation of solutions may be orders of magnitude slower than in research scenarios, which hinders training agents in environments that require a very large number of steps. In this work, we show that we can alleviate the computational burden of reward evaluation by actively sampling data points and learning a reward model. By using a cheap reward model, we can provide rewards much faster and, in addition, avoid repetitive and thus redundant evaluation of frequently visited states. We show that we can use the original *MolDQN* with an actively learned reward model to optimize properties of molecules which are much more costly to evaluate. We also show that we can train agents for hundreds of thousands of steps without excessive amounts of computational effort in an airfoil optimization task. This does not only contribute to *Green AI* (Schwartz et al., 2020) in these scenarios, but also may allow smaller institutions to engage in this area of research.

## 3  Our method: ACRL

Existing literature covers how to learn a reward function in cases of unclear tasks or how to make efficient use of it in cases where it exists and can be evaluated frequently. In contrast to that, in many other tasks the reward function is clearly defined but costly to evaluate. Providing these kinds of rewards to an agent during training thus can become prohibitively expensive even with off-policy learning with experience replay as one may fail to gather enough examples to learn from. In the following we describe our proposed ACRL framework to alleviate this issue. We use a standard MDP formulation as found in Sutton & Barto (2018).

Let $f(s)$ be a quantity or metric associated with state $s$, $f$ being a known but expensive to evaluate function of $s$. Without loss of generality, we aim to find a (local) minimum of $f$, or equivalently, a (locally) optimal state $s^* = \arg\min_s f(s)$. Due to high computational cost as well as non-convexity of $f$ in real-world tasks, we neither can directly solve for $s^*$ nor is it likely that we can find $s^*$ with heuristic search in general. We therefore propose a more principled search of $s^*$ by framing it as a sequential decision-making problem within the RL framework. A natural definition of reward in such environments is $r_t = f(s_{t-1}) - f(s_t)$, i.e., the agent aims to accumulate reward by sequentially visiting states $s$ with decreasing $f(s)$. We note that this formulation lends itself well to attract the agent to minima and is a popular choice in optimization scenarios (Khalil et al., 2017). Nevertheless, the standard cumulative, discounted return formulation can be whenever appropriate. Let $s_0$ be a possibly random initial state, the agent then aims to maximize the total cumulative reward $R_T = \sum_{t=1}^{T} f(s_{t-1}) - f(s_t) = f(s_0) - f(s_T)$. Due to the computational complexity of $f$, training an agent for a large number of steps may become infeasible or at least very time-consuming.

To reduce the computational burden of state evaluations during training, our framework requires only a few modifications of the standard training loop, namely the introduction of a reward model and its improvement via active learning. In general, the interaction of policy and reward model closely mirrors the interaction of policy and value networks in generalized policy iteration (Sutton & Barto, 2018), except that in our framework trajectories generated by the policy are used to improve the reward model, which itself is used to improve the value function. The two steps are then as follows.

The first step is to pre-train an approximate reward model $\hat{f}$, e.g., a neural network, on a small, initial dataset $D$ in a supervised manner. $\hat{f}$ is then used as a drop-in replacement for the true evaluation function $f$. Doing so is theoretically sound as the reward distribution does not depend on the agent's policy. This allows using our framework with both value-based and policy gradient methods without the necessity to change the underlying theory. At this point, we make several mild assumptions about $f$. In contrast to the general RL setting, we assume that we can evaluate $f$ in any state, thus providing dense and instantaneous rewards on state transitions. Hence, our method is not well-suited for sparse or delayed rewards, as found in conventional reinforcement learning scenarios, which we do not aim to cover since it is rarely the case that we cannot evaluate any property of a system.

The second step is then to actively improve the reward model during agent training. Since the initial state distribution in $D$ likely differs from states visited by an exploring agent, $\hat{f}$ may have poor extrapolation capabilities which will cause agent training to diverge as estimated state quantities may not have their true value predicted accurately. This particularly applies in scenarios where it is difficult to define *good* initial states, for example in the case of optimization problems where the optimal solution is to be found rather than given. To overcome this issue, we propose to sample a small number of states encountered during agent training and to provide the expensive ground-truth labels for them. In the most general form, we define an acquisition function $h(s)$ which hypothesizes about how beneficial adding the true label $f(s)$ to $D$ is for training the reward model. We then periodically evaluate $h$ for a small fraction of the agent's experience $\mathcal{E}$, e.g., the last $N$ steps, where $N$ is an application-dependent hyperparameter. We set $s' = \arg\max_{s \in \mathcal{E}} h(s)$, $D = D \cup \{s'\}$ and subsequently update $\hat{f}$ on the new $D$, either by training from scratch or fine-tuning. At this point, we assume that reward model can be trained reasonably fast such that the training time can be amortized given enough reward evaluations. For example, $h(s)$ may be chosen to be $\hat{f}(s)$, $||\nabla \hat{f}(s)||$ or other sampling techniques like uniform or uncertainty sampling. We hypothesize that this active learning component allows to explore relevant regions of the state space effectively and efficiently. An important implication of using active learning is that, depending on the task at hand, the initial reward model must not be perfectly accurate, which is a recurring theme in reinforcement learning when viewed from the perspective of policy iteration. For example, when using our method for optimization tasks where it is unlikely that the optimal solution space is included in the initial dataset $D$, perfect accuracy in this space is not necessary since the agent moves away from the initially covered space towards a more optimal region. It is thus more important for the reward model to be accurate on the on-policy distribution of states rather than on randomly selected initial data points. The reward model is only required to improve as the agent's policy improves and stabilizes. We found this active learning component to be crucial in our tasks.

A summary of the overall procedure can be found in Algorithm 1. We note that even though we use variations of Double DQN (Van Hasselt et al., 2016) agents in all experiments, our method does not assume any particular type of RL implementation and can be integrated into existing implementations with minimal changes, even in asynchronous and distributed settings. Due to the freedom of choice of algorithms, our framework can be used for both discrete and continuous optimization problems. The same holds for sampling strategies or active learning approaches in general.

## 4 Applications

### 4.1 Proof-of-principle: Molecular property optimization

The algorithm described above is first used in molecular property optimization tasks as a proof-of-principle. We use two fast-to-evaluate benchmarking properties to evaluate the performance of the algorithm and to choose its optimal hyperparameters. Both the Q-network and the reward network are trained on Morgan

---

**Algorithm 1** Double Deep-Q-Learning within ACRL

---

1: agent $A$, replay buffer $B$, initial dataset $D$, environment $E$, reward network $\hat{f}$ trained on $D$
2: $\hat{f} \leftarrow \text{train}(\hat{f}, D)$       ▷ train reward network
3: E.reward $\leftarrow \hat{f}$       ▷ E.step() uses $\hat{f}$ instead of $f$
4: **for** episode = 1 to M **do**       ▷ training loop
5:     $s_t \leftarrow$ initial state
6:     **for** step = 1 to T **do**       ▷ episode loop
7:         $a_t \leftarrow$ A.action($s_t$)       ▷ $\epsilon$-greedy
8:         $s_{t+1}, \hat{r}_{t+1} \leftarrow$ E.step($a_t$)       ▷ $\hat{r}_{t+1} = \hat{f}(s_t, a_t)$
9:         $obs \leftarrow (s_t, a_t, \hat{r}_{t+1}, s_{t+1})$
10:         B.add($obs$)       ▷ save observation
11:         $obs \leftarrow$ B.sample()       ▷ sample experience from B
12:         A.optimize($obs$)       ▷ update parameters
13:     **end for**
14:     **if** sample state **then**       ▷ e.g., periodically
15:         $s' \leftarrow \arg\max_{s \in \mathcal{E}} h(s)$       ▷ any method
16:         $y' \leftarrow f(s')$       ▷ calculate ground-truth label
17:         $D \leftarrow D \cup \{(s', y')\}$
18:     **end if**
19:     **if** update model **then**       ▷ e.g., periodically
20:         $\hat{f} \leftarrow \text{train}(\hat{f}, D)$       ▷ retrain reward network
21:         E.reward $\leftarrow \hat{f}$       ▷ update reward network
22:     **end if**
23: **end for**

---

fingerprint vectors as molecular representations (Morgan, 1965; Rogers & Hahn, 2010). States and actions are based on prior work (Zhou et al., 2019), where states are discrete molecular graphs and actions are semantically allowed local graph modifications.

The first benchmarking property is the penalized logP score, a widely used metric in the literature for evaluating and benchmarking machine learning models on regression and generative tasks (Nigam et al., 2020; Gómez-Bombarelli et al., 2018; You et al., 2018). The logP score is the logarithm of the water-octanol partition coefficient, quantifying the lipophilicity or hydrophobicity of a molecule. Penalized logP additionally takes into account the synthetic accessibility (SA) and the number of long cycles ($n_{cycles}$):

$$pen.\, logP = logP - SA - n_{cycles} \tag{1}$$

The second benchmarking property used here is the QED score, which is a quantitative estimate of druglikeness based on the concept of desirability (Bickerton et al., 2012). QED is an empirical score quantifying how "drug-like" a molecule is. Both properties are computationally inexpensive and can be calculated using RDKit (RdKit, 2006). We use them as benchmarking properties to study the effect of replacing the ground-truth reward with an approximation and to choose hyperparameters of our algorithm. In both applications, empty initial states are optimized for $T = 40$ steps. We then test our method on a real-life application in molecular improvement with a more costly property value to calculate.

## 4.2 Application I: Molecular design

In our first application, we evaluate ACRL on a molecular design task involving more costly rewards. We aim to optimize electronic properties of molecules such as energies of the Highest Occupied Molecular Orbital (HOMO) and the Lowest Unoccupied Molecular Orbital (LUMO) by performing sequential modifications. These values can be calculated using semiempirical quantum mechanical methods such as density functional tight binding methods as implemented in *xTB* (Grimme et al., 2017; Bannwarth et al., 2020). xTB-based reward evaluations on one Intel Xeon Gold 6248 CPU range from seconds to minutes, depending on size and structure of the

molecule. Compared to other RL applications, this is comparably expensive, especially considering the number of reward evaluations needed during agent training. The algorithm described above is applied using the hyperparameters found in the experiments of Section 4.1. Here, the agent learns a more application-oriented optimization goal, i.e., how to decrease the LUMO energy of randomly sampled starting molecules with only $T = 5$ steps per episode, while keeping the HOMO-LUMO gap constant. Therefore, the goal of the agent is to find optimal local improvements of given molecules with a limited number of actions, i.e., changes of the chemical structure. Let $s_0$ be a randomly sampled molecule at the beginning of an episode, then the improvement of the molecule $s_t$ at timestep $t$ over $s_0$ is defined as:

$$R(s_t) = -|\text{gap}(s_t) - \text{gap}(s_0)| - (\text{LUMO}(s_t) - \text{LUMO}(s_0)) \qquad (2)$$

with $\text{gap}(s) = \text{LUMO}(s) - \text{HOMO}(s)$ being the HOMO-LUMO energy difference of molecule $s$.

### 4.3   Application II: Optimization of airflow drag around an airfoil

The control technique of wall-normal blowing or/and suction constitutes a promising approach for the reduction of drag in turbulent boundary layers (Kinney, 1967). This technique has been successfully utilized not only in flat-plate boundary layers (Kametani & Fukagata, 2011) but also on more complex curved geometries like airfoils (Atzori et al., 2020). The majority of studies on the aforementioned control technique, however, considers uniform distribution of the introduced blowing or suction profiles. In our second application, we use ACRL to minimize aerodynamic drag around an airfoil by sequential adjustment of a set of blowing and suction coefficients represented as vectors in $\mathbb{R}^d$ (see figure 2a), which form the state space in $\mathbb{R}^{2d}$. As higher coefficients trivially reduce drag, we seek to optimize profiles with a constrained mean value for each side. By choosing a different constraint at the start of each episode, we aim to generalize across multiple instances of optimization. We use a Double DQN (Van Hasselt et al., 2016) agent with discrete actions corresponding to exactly one (or no) modification of an entry of $s$ per step to keep the action space as small as possible. Thus, we seek to find a (near-)optimal state $s^* \in \mathbb{R}^{2d}$ under given constraints. In our experiments, we use an episode length of $T = 30$ steps. While policy methods would be a more appropriate for this task, we use Double DQN for the sake of consistency.

Let $d_0 = f(s_0)$ be the drag coefficient of starting state $s_0$ corresponding to a uniform profile on each side. Our agent then seeks to find a sequence of modifications such that $R_T = \sum_{t=1}^{T} d_{t-1} - d_t = d_0 - d_T$ becomes as large as possible. We note that while the agent seeks to maximize $R_T$, we are primarily interested in the shape of states $s_T$ close to the (globally) optimal state $s^*$ rather than the exact value of $f(s_T)$.

The incompressible flow around airfoils is analysed using Reynolds-averaged Navier–Stokes equation based simulations in order to assess the effect of localized blowing and suction on the global aerodynamic performance of the airfoil. The simulations are carried out with the open-source CFD-toolbox OpenFOAM (Weller & Jasak, 2011) using a steady state, incompressible solver. For the current study we consider a flow around the NACA4412 airfoil at the Reynolds number $Re = U_\infty c/\nu = 4 \cdot 10^5$ and the angle of attack $\alpha = 5°$. For a more detailed description of the setup the reader is referred to Fahland et al. (2021).

One particular difficulty in training an RL agent in this scenario is the fact that the true state evaluation function $f$ is a Computational Fluid Dynamics (CFD) simulation. On one core of an Intel Xeon Platinum 8368 CPU, the simulation runs for approximately 10 minutes. Due to a fixed mesh size, we found that parallelization beyond 4 cores did not result in a significant speed-up, hence one reward evaluation takes approximately 2 to 3 minutes and cannot be reduced significantly, which severely limits the applicability of conventional RL algorithms with thousands of sequential reward evaluations.

## 5   Results and discussion

### 5.1   Molecular property optimization

Based on prior work by Zhou et al. (2019), we used cheap chemistry benchmarking properties logP and QED as a proof of concept to evaluate how the use of actively learned rewards performs in comparison to the real reward. Figure 1a and 1b show the performance of three different agents with NN-approximated

rewards compared to a reference agent ("oracle-based reward") trained on the real reward. One of the reward approximation agents is only trained once in the beginning ("static"). One of the agents ("ACLR") uses a reward model which is updated at regular intervals using additional oracle queries selected based on uncertainty sampling. The last agent ("full update") is updated after every episode using oracle queries of all states encountered in that episode (i.e., closest to the reference agent which directly uses oracle queries for training). After approximately 2000 episodes in case of logP optimization and already at the beginning of QED optimization, the performances of the agents start to differ. While the performance of the static agent stagnates, all three other agents show similar performance.

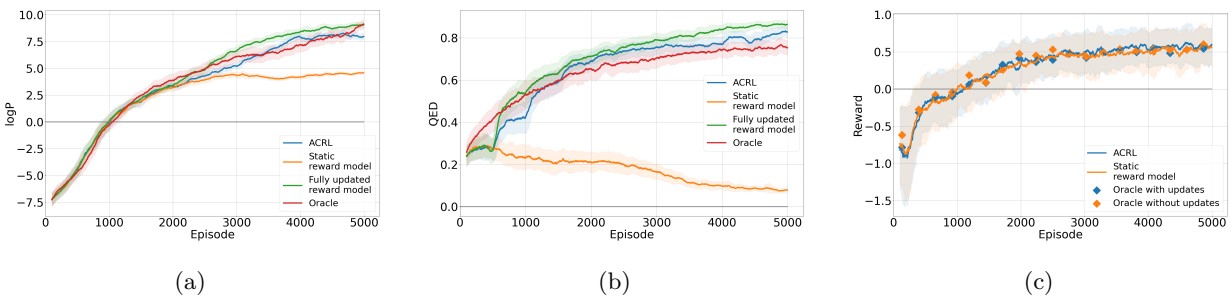

(a) (b) (c)

Figure 1: Evolution of the reward reached by the agent during the optimization of logP and QED: The red curve was obtained by training the agent on real (oracle-based) rewards, while the blue, orange and green curves are the ACRL model, the static reward model and a fully updated reward model, respectively. Due to high computational costs, only ACRL and static models can be tested in (c).

The failure of the **static agent** to learn is due to the low generalization ability of the initial reward model itself, which is trained on the QM9 dataset (Ruddigkeit et al., 2012; Ramakrishnan et al., 2014) containing approximately 134.000 molecules with up to 9 non-hydrogen atoms. To some extent, the weak generalization can be attributed to not using state-of-the-art graph neural networks. However, we decided to use the same molecular representation and model as in the original Q-networks in Zhou et al. (2019), i.e., fingerprint representations and MLPs. Furthermore, during the learning process, the molecules generated (especially after a high number of episodes) contain many more atoms, which explains why the static reward model fails to correctly estimate the real property values. The strength of this effect depends on the property studied. In the logP optimization task, the property values reached with a static reward model follow the general trend of learning with real reward, even though the final performance after 5000 episodes is lower. In case of QED optimization, the static reward model fails to predict QED-values for molecules outside the training distribution. As a consequence, the RL agent learns to exploit errors of the static reward model and finds adversarial examples, rather than samples with desirable properties.

The active learning component within **ACRL agent** allows the reward model to learn from molecules outside its initial training distribution, thus improving reward evaluations during agent training. By only selecting a small subset of labels obtained using oracle queries to be added to the training set, the objective of the ACLR agent is to mimic the reference agent's real behavior as closely as possible. This includes finding (nearly) optimal points (see e.g., Lindner et al. (2021)) to be selected for retraining of the reward model to minimize its errors while at the same time minimizing the number of costly oracle queries. We experimented with different sampling strategies (see SI), from which a query-of-committee model (see Seung et al. (1992)) performed best. Therefore, in the ACLR model used in the molecular design and improvement tasks, three reward models were trained independently to form a query-of-committee model. The three reward models are retrained after 500 episodes with the initial training set along with all 400 new molecules generated during the agent learning process and their computed real property values. The selection of new oracle queries to extend the dataset is based on the disagreement between the three reward models measured by the standard deviation of the predictions. However, our work is independent of the particular sampling strategy (even random sampling of visited states can work well in some applications), as long as the reward model's training distribution follows the exploration of the RL agent. Overall, the speed-up achieved by the ACRL model in

this experiment compared to the fully updated and oracle-based model is 50 (see Table 1). The relationship between speed-up and rewards reached is analyzed in the SI.

| Task | Oracle queries | Model queries | Relative speed-up | Oracle duration | Model duration | Oracle time | Model time |
|------|---------|---------|-------------|----------|----------|--------|-------|
| Mol. opt. | $\sim$4,000 | $\sim$200,000 | $\sim$50 | $\sim$1s | $\sim$0.001s | $\sim$55h | $\sim$0.05h |
| Mol. imp. | $\sim$4,000 | $\sim$25,000 | $\sim$6.25 | $\sim$1m | $\sim$0.001s | $\sim$7h | $\sim$0.007h |
| Drag opt. | $\sim$3,000 | $\sim$9,000,000 | $\sim$3,000 | $\sim$3m | $\sim$0.001s | $\sim$7500h | $\sim$2.5h |

Table 1: Relative and absolute speed-up factors for different tasks comparing the number of oracle and model queries. The duration of a reward model forward pass has been conservatively estimated as $1ms$. Wallclock durations are calculated based on the number of steps the agents trained. Note that the absolute times are only for evaluating rewards, not including training time. For the drag optimization task, the agent trained for roughly 10h. Training on ground-truth rewards would require lots of parallel agents to train in a reasonable timeframe, while we train only one agent at a time.

**Fully updating** the reward model on oracle queries of all samples ("full update") aids the learning process. In case of logP optimization and even stronger in case of QED optimization, learning by fully updating the reward model has even surpassed learning with actual reward values at certain episodes. One potential reason for that can be that the exploration of the fully updated agent is stronger than that of the reference agent (see SI), which needs to be confirmed in future work. However, in practice, fully updating the reward model by adding every single generated point (along with its real property value) to the initial dataset and retraining the neural network is as expensive as training the reference agent, so it cannot be applied to tasks with costly rewards.

In order to understand why the fully updated reward model in some cases (e.g., Figure 1b) outperforms the oracle-based training, we analyzed the effect of additional noise and thus exploration which might be induced by replacing oracle-based rewards with (noisy) approximated rewards. We therefore varied the $\epsilon$-greedy strategy of the learning process. In particular, we varied final $\epsilon$ values (i.e., probabilities of random actions) and the form of the $\epsilon$-decay function used on the learning process. However, none of the changes in $\epsilon$-decay could improve the learning behaviour, i.e., the $\epsilon$-decay rate and function used by Zhou et al. (2019) was optimal. Therefore, for the rest of the simulations we used a fully exponential decay reaching approximately 1% randomness in episode 5000. The results of this study are available in the supplementary information section. Further study of the improvement effect due to a fully updated reward model is part of ongoing work as it has the potential to improve the performance of RL agents with little computational overhead.

### 5.2 Molecular improvement

After evaluating the performance of our agent on easy-to-compute properties such as penalized logP and QED, we test our ACRL approach on a molecular improvement task with more costly rewards, where an oracle-based reference study is unfeasible. In particular, we study a RL agent with the goal of independently varying two quantum mechanically calculated energy levels of molecules with only very few, in our case five, modification steps (see Section 4). Figure 1c shows the evolution of the ACRL and the static reward agents' rewards as a function of the training episode. We observe that the reward becomes positive after approximately 1000 episodes and stagnates after approximately 2000 episodes. Therefore, the agent has learned to improve given (arbitrary) molecules, since the reward value of the starting reference molecule is zero, each episode starts with a randomly sampled molecule, and any molecule with negative reward would have less desirable properties than the initial one. This suggests that even though the agent deals with different starting reference molecules at each episode, it has managed to learn a strategy to increase the reward in a limited number of steps.

In contrast to the property optimization task discussed before, the performance of the ACRL and the static reward agents are equal within the confidence intervals. A likely explanation for this observation is that the number of steps per epoch in this task is limited to five, whereas 40 steps were possible in the prior task. Therefore, the agent here cannot generate molecules that are far outside the initial distribution of

starting molecules, i.e., the QM9 dataset. Furthermore, the ratio of reward model queries to oracle queries in this experiment is comparably high (see Table 1), meaning that the ACRL reward model is updated on a high fraction of actually encountered molecules. A reference calculation with oracle based rewards or a fully updated reward model to check if the ACRL model found near-optimal results (within the DQN framework) are computationally too costly here and thus unfeasible. However, we compared the predictions of the reward models for randomly selected molecules throughout the training process to oracle predictions (see points in Figure 1c). We found excellent agreement, indicating that the ACRL as well as the static reward models are reliable. Thus, the solutions found are not exploiting weaknesses of the reward models, nor is the training limited by wrong predictions of the reward models. Therefore, it is likely that the found solutions are of comparable quality as ones that a hypothetical oracle-based RL model would find. The speed-up achieved in this experiment compared to a hypothetical oracle based model is 6.25, which still has room for improvement, given the high reliability of the reward models. Even though the ACRL agent does not exceed the performance of a static reward model trained on QM9, these results have another important implication. While QM9 contains all molecules with up to nine heavy atoms, of which around 130,000 exist and on which the static model has been trained on, the ACRL agent matches its performance using only 4000 ground-truth queries. While large databases exist for molecules, this may not be the case in other and especially new domains. This showcases the effectiveness of our approach in the low-data regime.

### 5.3 Optimization of airflow drag around an airfoil

Our ACRL method is applicable to a large number of different tasks in natural sciences and engineering, not only limited to chemistry. Therefore, in this section we present the results of a task in engineering, namely the reduction of airflow drag around an airfoil, e.g. an airplane wing (see Section 4). The objective in this task was to find a set of coefficients minimizing drag and to analyze the resulting profiles. Figure 2b shows the evolution of drag during 300000 episodes of training. The discrete jumps of the ACRL model coincide with retraining of the reward model every 10000 episodes. As higher mean constraints are highly correlated with lower drag, we choose samples for ground-truth evaluation based on reward rather than drag. Dots represent oracle-based ground-truth evaluations of random profiles sampled during training.

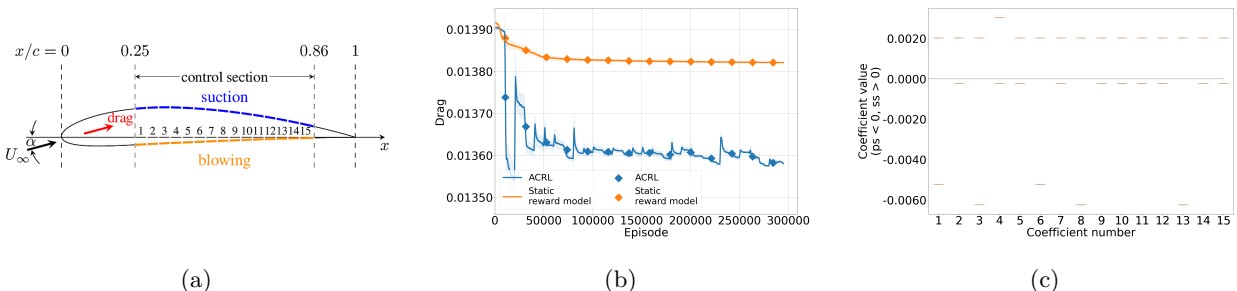

Figure 2: (a) blowing/suction distribution discretized with 30 coefficients corresponding to 15 sections on each side of the considered airfoil. (b) drag evolution of two independent runs. (c) coefficient distribution for low-drag profiles.

The results demonstrate that the ACRL agent is able to find profiles with significantly lower drag coefficients than the static reward model. They also show that in this task (in contrast to the molecular improvement task) it is crucial to actively update the reward model during training. This is related to the fact that in order to improve upon the initially uniform profile, the RL agent has to perform a constrained optimization in high-dimensional real space (30-dimensional in our case). Accurate reward model predictions require sufficient coverage of the relevant space within the initial dataset which is difficult to assert because the relevant region is, in general, not known, which is also true for many other real-world problems. As a consequence, an agent trained without active updates of the reward model only slightly improves upon a uniform profile. At the same time, model updates result in sharp drops of both predicted and ground-truth drag especially in the beginning of training as the relative effect of new ground-truth samples is high and the RL agent probably

exploits wrong predictions of the early-stage reward models. This effect decreases as more and more samples are obtained along the trajectories towards the low-drag region in parameter space.

Figure 2c shows the distribution of a small number of low-drag profiles sampled with ground-truth labels during training. The resulting profiles are non-trivial and have a regular, alternating pattern of coefficients with physically explainable meaning (Kametani et al., 2016; Mahfoze et al., 2019; Stroh et al., 2016). We note that due to various limitations of the simulated environment such as discretization of action space, a limited number of coefficients due to limitations in OpenFOAM simulations (used as an oracle) and limited episode length, these results are only locally optimal w.r.t. our setup. Yet, we find consistent, physically interpretable and highly non-trivial results.

## 6 Conclusion

We introduced ACRL, an extension to standard reinforcement learning methods in the context of (computationally) expensive rewards, which models the reward of given applications using machine learning models. Because optimal regions in the search spaces are not known a priori and thus typically are not included in initial training sets, we use active learning while exploring the state space to update the reward model over the course of training. We first showed that it is possible to train agents with an incrementally improving reward model on existing benchmark tasks using cheap benchmark quantities. We then showed in two more realistic scenarios that by learning a reward model jointly with our policy, we can reduce the time for reward evaluations by several orders while still being able to produce meaningful results. In turn, it becomes feasible to train agents without massive distribution within reasonable timeframes, which saves computational resources and energy and at the same time accelerates research since resources can be spent on training models rather than evaluating rewards.

**Acknowledgements**
We would like to thank the Federal Ministry of Economics and Energy under Grant No. KK5139001AP0. We acknowledge support by the Federal Ministry of Education and Research (BMBF) Grant No. 01DM21001B (German-Canadian Materials Acceleration Center). We acknowledge funding by the German Research Foundation (Deutsche Forschungsgemeinschaft, DFG) within Priority Programme SPP 2331.

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

## 7  Supplementary Information

### Comparison of querying strategies for retraining

The three strategies for selection of points presented were compared on a constant number of selected points (800) (Figure 3). In each case, three reward models are initially trained with three different train-test splits of the original QM9 dataset and used for prediction later on. The mean of the three reward models' predictions is used as a final reward for the agent to maximize, and the standard deviation of these are calculated to give an idea of the uncertainty on prediction. Based on these models, three selection modes are studied. In the first setting, the models are retrained by randomly sampling a number of points from the initial QM9 test set as well as newly generated points during the learning process. In a second setting, the points with the highest standard deviations of model predictions are sampled and the models are updated using these points as train set. This is based on the assumption that the points with the highest standard deviations of model predictions (points in which the three models "disagree") are more likely to come from outside the original train distribution, thus potentially representing the points that the model needs to learn from in order to improve its predictions during the agent training process. In this sense, having three reward models instead of only one could provide a good basis for the selection of points. Finally, a third approach in sampling points is based on classifying the previously obtained test set and newly generated points in different bins before selecting points with the highest standard deviation (of model predictions). This stems from the fact that in certain situations, points with the highest standard deviations could represent outliers, and therefore could prevent the reward models from learning the main trend of the data. The strategy of bin-based selection offers a solution to this by selecting points with the highest standard deviations while respecting the initial data distribution. It is important to note that all these sampling processes are done separately for each model to be retrained, since their respective starting train and test sets are not the same to start with. Therefore, the three models will never be updated on the same training data, and will thus provide independent predictions, depending on the points they were trained on. Overall, the standard deviation based sampling method performed best and was thus used in the main part of this paper.

### Comparison of the number of points selected for retraining

Given the best strategy (standard deviation based), the effect of the number of points selected for reward model retraining was studied (Figure 4). The minimal number of points that was consistently comparable to the real reward was 400 points.

### Study of the effect of varying degrees of randomness on learning

The $\epsilon$ values in the MolDQN paper start with values of 100% and decrease exponentially at each episode until they reach a percentage of 1% at episode 5000. In this study, our aim was to compare the effect of the $\epsilon$ end value (last value of randomness) on the real reward agent at episode 4800, considering that the agent

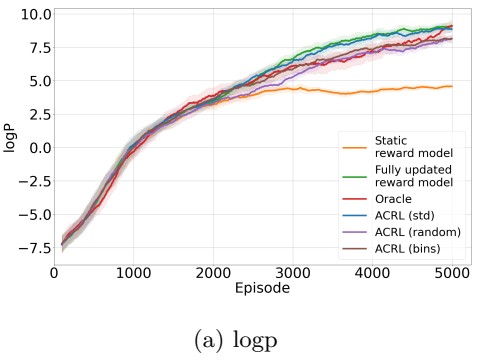

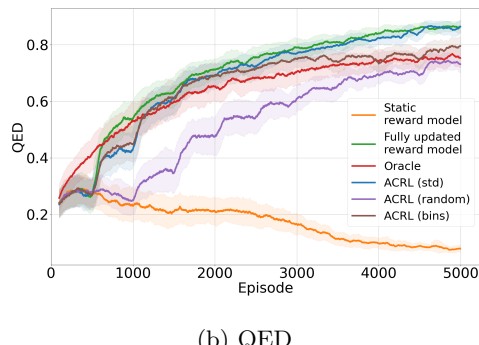

|  (a) logp  |  (b) QED  |

Figure 3: Comparison of different selection modes, random (purple), standard deviation based (blue) and bin-based (brown) on 800 sampled points.

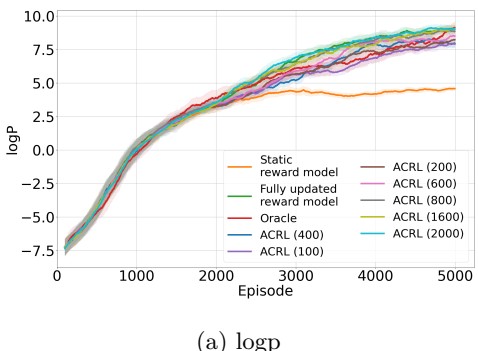

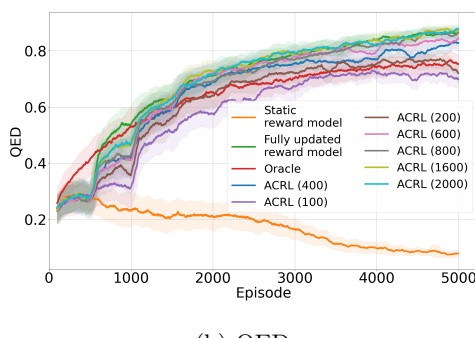

|  (a) logp  |  (b) QED  |

Figure 4: Comparison of reward models retrained on a varying number of points selected based on standard deviation

does not choose any more random actions from episode 4800 to 5000 (to better guide it towards the end goal). Results are shown in Figure 5. We concluded that increasing the $\epsilon$ end values (increasing randomness, with the hope of favoring exploration) did not help the agent reach better rewards.

**Study of the effect of the decay function form**

The $\epsilon$-decay function used in MolDQN is an exponential decay function. In this study, we choose to study the effect of varying decay function forms by adding a linear component to the exponential function with varying fractions. The equation is the following:

$$\epsilon(t) = \epsilon_0(\lambda(1 - \beta t) + (1 - \lambda)\alpha^t) \tag{3}$$

with $1 - \beta t$ and $\alpha^t$ the linear and exponential components respectively, $\epsilon_0$ the starting randomness (at 100%), $\beta$ and $\alpha$ constants that depend on the starting and end values (we choose an end value of 1% at episode 4800), and $\lambda$ the fraction (relative importance) of the linear component. At a $\lambda$ of 0, the decay-function is fully exponential, and at a $\lambda$ of 1, it is fully linear. Results are shown in Figure 6. We conclude here that a fully exponential function is more convenient for the learning of the agent.

**Comparison of different mean constraints**

In this section, we present a more detailed description of the results for the drag optimization task. Our initial training dataset contains 5000 random profiles with a mean value centered around $\pm 0.002$ on each side. For a constraint configuration matching the distribution of the initial dataset, Figures 7a and 7b show the evolution of drag and reward, respectively. In order to test extrapolation and generalization capabilities of

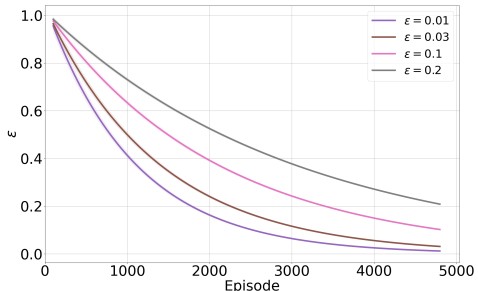

(a) $\epsilon$-values' exponential decay with varying end points

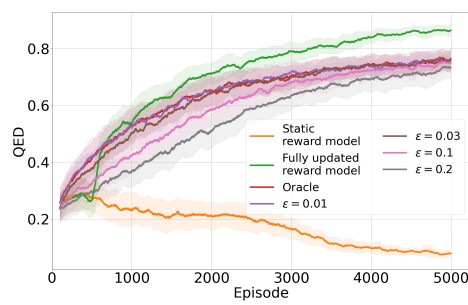

(b) Results on the QED task

Figure 5: The effect of increasing randomness by reaching different $\epsilon$ end values with the same exponential decay function.

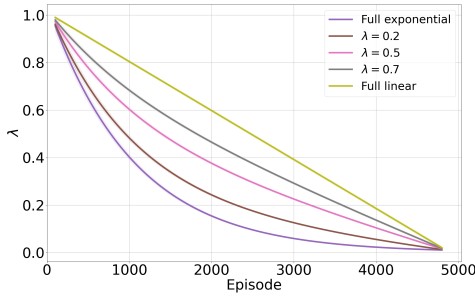

(a) Different forms of $\epsilon$-decay functions

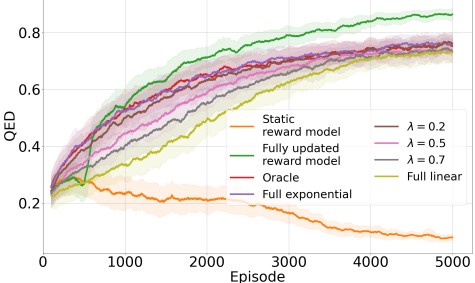

(b) Results on the QED task

Figure 6: The effect of different $\epsilon$-decay functions with the same randomness end value.

our ACRL method, we repeated the same experiment with a larger constraint interval which lies outside the initial training distribution. The results in Figures 8a and 8b show that ACRL is able to explore the underrepresented space well, in contrast to a static reward model which fails to guide the agent to explore the relevant solution spaces. The higher variance stems from the fact that higher mean values (or equivalently, higher total volume) trivially reduce drag. Thus, in this experiment the agent encounters a higher diversity of states in terms of their constraint. Even though most of the observed states lie outside the initial training distribution, an ACRL agent is still able to explore the relevant low-drag space. The importance of actively updating the reward model during training is reflected by the results of agents using a static reward model in both experiments. Both agents trained with a static reward model achieve very similar results in terms of drag, even though drag distributions vary considerably between the experiments. Only the ACRL agents are able to capture the variance of drag well, which is especially high in Figure 8 due a larger constraint interval. In contrast, static models fail to move outside their initially modelled distribution, even though they predict drag values of encountered states very accurately.

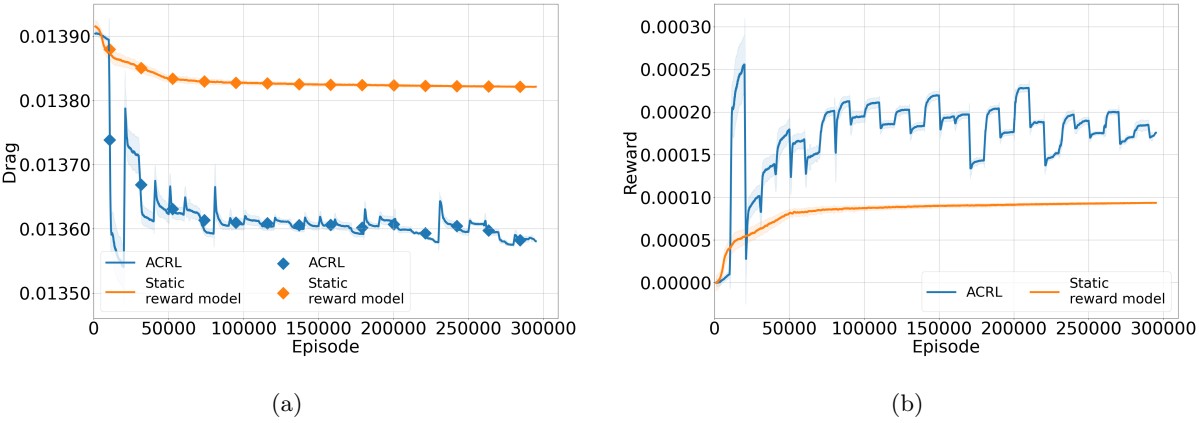

Figure 7: Results for different mean constraints in $[0.0019, 0.0021]$.

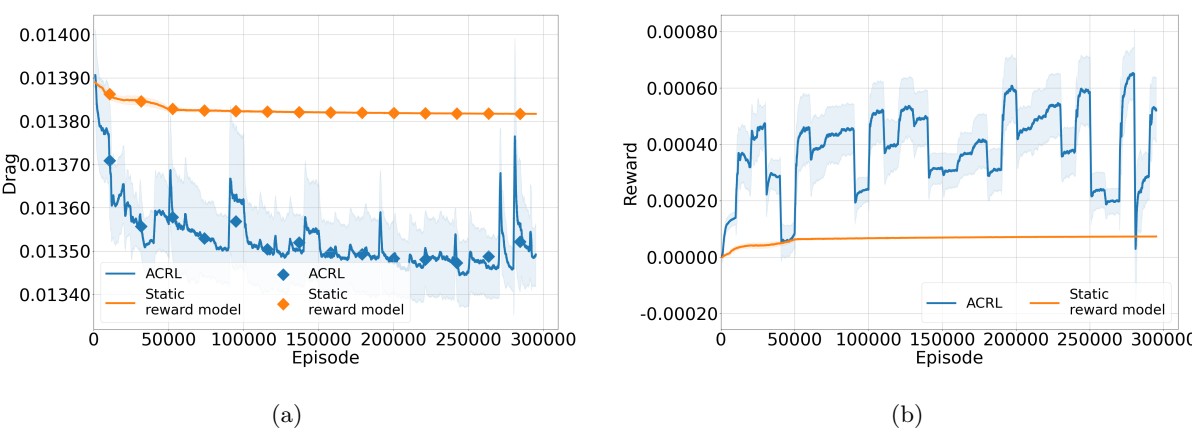

Figure 8: Results for different mean constraints in $[0.0015, 0.0025]$.

