# OpenReview forum: "Actively Learning Costly Reward Functions for Reinforcement Learning"
_TMLR — Rejected by TMLR_

### Review · Reviewer_mm74 · 2023-05-08

**Summary Of Contributions:**

The authors propose a new deep reinforcement learning method suitable for a wide range of applications in science and engineering. The method called actively learning costly rewards for reinforcement learning (ACRL), provides a fast and efficient way to solve optimization problems with a known ground-truth reward function. Evaluating the ground-truth reward function can be expensive for many science and engineering problems.
ACRL involves the following key steps: pre-training a neural network model to approximate the ground truth reward function and then using active learning to update the model during training. The authors demonstrate the effectiveness of ACRL with double deep Q-learning on three use cases: molecular property optimization, molecular design, and optimization of airflow drag around an airfoil. The first two use cases have a known optimization function, while the third relies on querying from CFD simulations.
Overall, this work presents an interesting approach to engineering optimization problems and showcases that most optimization problems can be framed as RL problems and approximated by deep learning models.


**Audience:**

Yes

**Claims And Evidence:**

Yes

**Requested Changes:**

This work would be further improved if the authors addressed the following questions:

•	As mentioned above, the authors should review relevant works in traditional surrogate modeling approaches (e.g. Engineering Design via Surrogate Modelling: A Practical Guide, DOI:10.1002/9780470770801) and Bayesian optimization methods (e.g. https://pubs.acs.org/doi/abs/10.1021/acs.jcim.1c00637, https://www.nature.com/articles/s41586-021-03213-y) in the introduction section. They should also point out the key differences between ACRL and these methods.

•	The authors state that the ground truth reward functions have to be known, but later indicate that the CFD simulation is one of the ground truth reward functions. The word "known" is ambiguous. Does "known" mean that the mathematical model has to be known, or can it simply be queried as is the case for CFD? These are two different concepts. Namely, is the author referring to a "black box function" or a "white box function"?

•	On a similar note, the authors state that "IDRL assumes the absence of a reward function". Isn't this the more common case in science and engineering where the reward function is unknown ("black box function")?

•	The authors should explain how to construct the initial set of samples and training. Is there a strategy to sample (random or uniform)? How does the active learning approach balance the exploration and exploitation trade-off? When to choose uniform and when to choose uncertainty sampling? In section 5.1, 500 episodes are used for initial training. The authors didn't mention the number of initial samples for the latter two use cases in sections 5.2 and 5.3. These are key questions in active learning that need to be addressed.


**Strengths And Weaknesses:**

The strengths of this work include:
•	The paper is well-written, and the language is easy to understand.

•	The authors discuss the key differences between this work and prior works in RL, such as IDRL, which highlights the uniqueness and progress made in this work.

•	The ACRL method is presented well in mathematical terms, and the algorithm is easy to understand.

•	For each use case, the authors provide detailed background information on the optimization problems, making this work accessible to readers with little or no knowledge of chemistry or engineering.

The weaknesses of this work include:
•	The authors did not compare ACRL with methods often used in science and engineering where the original optimization function is expensive to evaluate. These methods include surrogate modeling approaches and Bayesian Optimization. This raises questions about the level of innovation in this work, as DQN is just another surrogate model and the optimization problem is framed as RL.

•	The manuscript is missing many technical details of the active learning approach, such as the choice of initial sampling strategy, acquisition function, and how to balance the exploration and exploitation trade-off.

---

### Review · Reviewer_dKRh · 2023-05-12

**Summary Of Contributions:**

The authors propose actively learning costly rewards for reinforcement learning (ACRL). This particular method works in settings where a reward function is known but hard to compute and a learned reward model can improve wall-clock time. The authors focus on problems that are not standard RL benchmarks: chemistry, materials science and engineering. The authors provide empirical results on toy examples as well as more practical problems to demonstrate its effectiveness.

**Audience:**

Yes

**Claims And Evidence:**

Yes

**Requested Changes:**

- How does the proposed method compare to related works (on using a learned reward model) that can be applied to the same problems, or can be applied with some small adjustments? If all previous works are absolutely impossible to apply to the problems studied in this paper, the authors should emphasize that in the paper (maybe using a table like the Table 1 in IDRL paper to emphasize it), if they can be somehow applied, then there should be some comparison experiments. (it is OK to show, for example, an alternative solution will give similar performance but takes much longer wall-clock time, but there should be some comparison).


Other minor changes:

- Please add additional explanations on what is (c in figure 1 in its caption to help reading. You mentioned in later section, but it’s good to also mention it in the caption.


**Strengths And Weaknesses:**

Strengths:

- **Clarity**: the paper is easy to read and is overall clear
- **Presentation**: overall structure, figures, and tables are clear
- **Significance**: the proposed method seems to be effective in reducing wall-clock time, and the results seem promising. The results can be quite interesting to people who work in these scientific problems and want to use a faster RL solution. And these problems are important problems but are less studied in the RL literature, which might add to the significance.

Weaknesses:

- My only major concern is on **Related work**: the authors discussed how the work is different from previous works and emphasized the differences, which is good. And the authors provided ablations on the proposed method which is also good. However, I am a little concerned about the fact that the proposed method is not compared to **any** other alternative methods. For example, authors mentioned IDRL and how it is different from the proposed method, but does it not work at all in the problems studied in this paper (or, maybe even a naive adaptation of IDRL to the problems in this paper, if IDRL has not been tested on the same problems). While the proposed method seems to be quite good compared to a naive baseline, it is unclear to me how it compares to related works in the same research direction.

Overall seems a good paper, I am happy to discuss the concern with authors and if I missed sth please point it out.

---

### Review · Reviewer_KkHV · 2023-05-13

**Summary Of Contributions:**

The authors present a method called ACRL to actively learn a reward function $f'$ when a real reward function $f$ exists but is very expensive to evaluate. The idea of ACRL is to train a neural network to approximate $f$ using a small set of samples obtained by evaluating $f$. The authors propose 3 strategies to build the training set for $f'$. The performance of ACRL is evaluated on chemistry and physics data sets.



**Audience:**

Yes

**Claims And Evidence:**

No

**Requested Changes:**

* Major changes
    * If the claim is to reduce wall time, then the wall time needs to be reported in experiments.
    * I also encourage to do an experiment that shows ACRL can achieve higher rewards with fewer samples than a reward model trained with random samples.
    * I think the strategy of incrementally build a training set is very important to active learning. But those strategies are only described in the supplementary. I would suggest move it to the main paper.
    * Clarify why number of steps are set differently for different tasks in the experiment.
    * Figure 1 (c) is confusing because it shows that ACRL has no advantage. I would suggest using a positive example instead of explaining why it did not work because it is supposed to support the main claim of the paper.
* Minor changes
    * In page 8, 'the ratio of reward model queries to oracle queries in this experiment is comparably low', should it be high?
    * In Figure 3, the caption and legend do not agree.
    * In Figure 8b, ACRL seems not converged.


**Strengths And Weaknesses:**

Strengths
* The problem is well motivated
* The paper is generally well written and easy to follow

Weaknesses
* In Section 2.2, the authors stress that the proposed method is more efficient w.r.t. wall time. However, there is not a single wall time comparison in the experiment.
* The speed-up shown in Table 1 is not convincing. I don't understand why using more queries means more efficiency. If cheaper querying time is the point, we still need to know the total training time to make that arguement.
* The experiment does not show if active learning reduces sample complexity.
* No baseline comparison. Some related works are mentioned in Sec 2.2 but none of them was compared with. They may have slightly different assumptions, but they can be adapted fairly easily.
* Some experiment settings need more explanation. E.g., why is a very short horizon $(T=5)$ used in the molecular design task, whereas much longer horizons are used in other tasks.

---

### Review · Reviewer_YpHc · 2023-05-25

**Summary Of Contributions:**

The paper presents a method of learning reward functions, which can be useful when evaluating of the reward function is costly. In a nutshell, the algorithm clones a (costly) ground truth reward function, into (cheaper-to-evaluate) neural network. It periodically retrains the neural network to mitigate the modelling errors on the previously unseen parts of the state space.

The algorithm is evaluated on three domains, and is used to search for local minima of functions related to molecular optimization, molecular design and airflow drug.

**Audience:**

Yes

**Claims And Evidence:**

No

**Requested Changes:**

I would like to see some comments addressing the weaknesses specified above.

Ad 1/4. Perhaps it would be better to state explicitly what is the main contribution. If it is ACRL, as it seems at the moment, please provide more environments. I'd be great, it they would cover some standard RL tasks.
Ad 2. Please provide the action space.
Ad 4. Please motivate why to use RL instead of optimization techniques (designed specifically to search for local minima).

**Strengths And Weaknesses:**

Strengths:
 - a straightforward method (easy to implement and control)

Weaknesses:
1. lack of focus about contributions - I am confused by what is the main contribution of the paper. Is it the ACRL method. If yes, that it should be benchmarked more properly, on more tasks/setups etc, and perhaps the detailed description of environments presented in the experimental section should be deferred to the appendix. At the moment, it feels more like the paper is focused on solving these particular tasks.
2. lack of clarity - the paper casts optimization problem into the RL setting, but does not specify the action space (in Sec 4.1/Sec 4.2)
3. non-standard RL setup - the paper claims: "we use a standard MDP formulation ...", however, it restricts to the optimization problems with a specific rewards, given by the difference of the potential
4. optimization with RL - RL is perhaps not the best tool for looking for local minima. At the very least, there should be some motivation provided, why it might be the case. Similarly, the baseline, in this case, should be 'standard' optimization techniques. Why, for example, SGD would not work?

---

### Decision · Action_Editors · 2023-06-18

**Recommendation:** Reject

**Comment:**

Due to lack of baselines the paper both lacks proper support for the main claim, as well as the main claim is too narrow (due to being restricted to methods developed in the context of reinforcement learning field, while not being clear on why one cannot use methods that wouldn't be described usually as belonging to the field).

**Audience:**

The paper is well-written and easy to understand (as emphasized by dKRh, mm74, KkHV). Most reviewers also agree that the problem is also well-motivated and is of great importance to many fields.

However, a major concern raised by **all** four reviewers is that the authors did not compare the proposed ACRL method with other existing alternative methods outside the field of reinforcement learning. The challenge of costly to evaluate reward is very commonly faced in many fields, and a vast array of solutions is available (e.g. Bayesian Optimization). As such the proposed paper would not be of interest to the broader community, as it is too narrowly focused on reinforcement learning.

**Claims And Evidence:**

The authors propose Actively Learning Costly Rewards for Reinforcement Learning (ACRL), in which the Authors replace costly reward function with a learned model. The main claim is that the proposed algorithm enables learning much faster than possible before using methods used in the context of reinforcement learning algorithms.

The Authors did not compare against RL baselines other than vanilla reinforcement learning such as IDRL (see dKRh comment).